# VISUO-TACTILE WORLD MODELS

## ABSTRACT

We introduce multi-task Visuo-Tactile World Models (VT-WM), which capture the physics of contact through touch reasoning. By complementing vision with tactile images, VT-WM better understands robot–object interactions in contact-rich tasks, avoiding common failure modes of vision-only models under occlusion or ambiguous contact states, such as objects disappearing, teleporting, or moving in ways that violate basic physics. Trained across a set of contact-rich manipulation tasks, VT-WM improves physical fidelity in imagination, achieving 33% better performance at maintaining object permanence and 29% better compliance with the laws of motion in autoregressive rollouts. Moreover, experiments show that grounding in contact dynamics also translates to planning. In zero-shot real-robot experiments, VT-WM achieves up to 35% higher success rates, with the largest gains in multi-step, contact-rich tasks. Finally, VT-WM shows data efficiency when targeting a new task, outperforming a behavioral cloning policy by over $3.5\times$ in success rate with limited demonstrations.

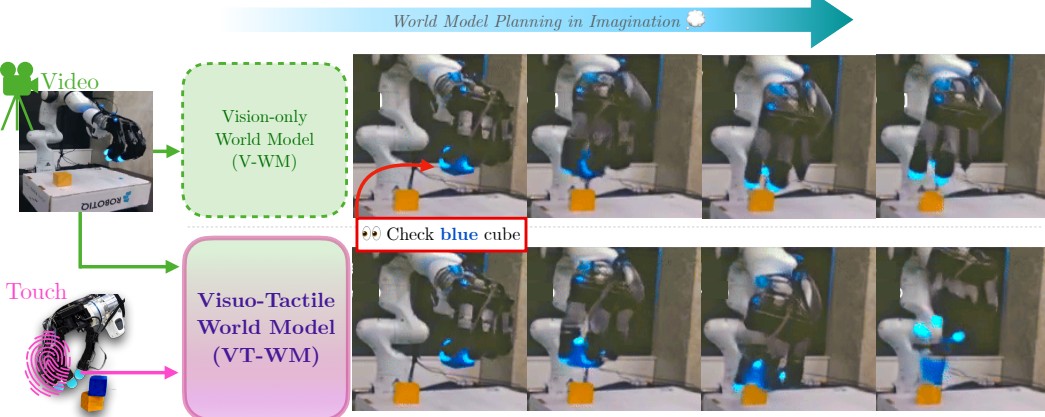

Figure 1: Visuo-Tactile World Model complements vision with touch, providing contact grounding of robot-object interactions. Notice that when using the WMs for planning a cube stacking task, the VT-WM has notion of object permanence of the **blue cube** when transporting, placing and releasing the object. The contact grounding provided by the vision-based tactile sensor helps to reduce hallucinations often present in V-WMs, enabling more reliable zero-shot planning in contact-rich manipulation tasks.

## 1 INTRODUCTION

World models (WMs) have emerged as a leading paradigm in machine learning, offering robots the ability to understand the physical world and plan interactions in *imagination* (Agarwal et al., 2025; Russell et al., 2025; Liao et al., 2025). In this work, we advance world models for robot manipulation by extending beyond purely visual imagination to incorporate modalities that directly ground contact interactions (fig. 1). By complementing vision with touch, world models for robot manipulation gain access to local contact signals that anchor predictions in the physics of contact. Tactile sensing provides this crucial information, enabling the model to capture object permanence and force-driven motion, and to move beyond the ambiguities and aliasing of vision alone.

We introduce the first multi-task `Visuo-Tactile World Model` (VT-WM). Vision provides global context about the robot's kinematics and the task scene, but it does not reveal the state of

physical contact. Tactile sensing supplies this missing local signal, capturing how the hand and object actually interact. Together, these modalities enable the model to maintain object permanence even under heavy occlusion or visual aliasing. As shown in fig. 1, VT-WM consistently represents the cube throughout the phases of a stacking task: in-hand during transport and placement, and back in the scene once released. Multimodality also disambiguates visually similar states with different outcomes. For example, from the camera view the robot hand may appear to rest on a cloth, yet only tactile feedback can reveal whether contact is sufficient for the cloth to move when wiping, or if it will remain in place. By grounding imagination in both global vision and local touch, VT-WM preserves object permanence and predicts object–robot interactions that respect the laws of motion.

We evaluate the gains of VT-WM over V-WM on a set of contact-rich manipulation tasks. Our first focus is how visuo-tactile training improves imagination by preserving object permanence and adhering to physical motion laws during autoregressive rollouts. Tactile grounding helps prevent common hallucinations in vision-only models, such as objects disappearing under occlusion, teleporting, or moving without applied forces due to visual aliasing. We then assess how this grounding translates to planning. While V-WM and VT-WM perform similarly on reaching tasks that mainly test kinematic fidelity, zero-shot plans generated by VT-WM show a stronger ability to maintain contact in the real world. This capability proves crucial for manipulation actions such as pushing, wiping, and placing, where reliable hand-object interaction determines task success.

Our contributions are threefold:

- We propose the first multi-task visuo-tactile world model that integrates fingertip tactile sensing with vision to jointly model global context and local contact dynamics.
- We show that visuo-tactile grounding substantially improves imagination quality, achieving a 33% gain in object permanence and a 29% gain in compliance with physical laws, evaluated across a set of manipulation tasks.
- We demonstrate that these improvements in imagination enable more reliable zero-shot planning on real robots, with up to 35% higher success in contact-rich tasks.

## 2 RELATED WORKS

In this work we aim to train general (multi-task) world models that can leverage both tactile and visual observations. Specifically we train latent-state world models, which first project observations into latent representations and then train an action-conditioned dynamics model in that latent space.

**Foundational encoders for vision and touch:** Unlike the established hardware platforms for computer vision, the field of robotic tactile sensing lacks a single standardized sensor. Nevertheless, vision-based tactile sensors have emerged as one of the most prominent solutions. Devices like GelSight (Yuan et al., 2017), Digit (Lambeta et al., 2020), and the more recent Digit 360 (Lambeta et al., 2024) capture tactile information by imaging the deformation of a soft elastomer surface.

Similar to the development of general-purpose visual encoders like CLIP (Radford et al., 2021), DINO (Caron et al., 2021; Oquab et al., 2023), I-JEPA (Assran et al., 2023), and Cosmos Tokenizer (Agarwal et al., 2025), recent efforts have focused on creating foundational models for vision-based tactile sensors through self-supervised learning. Models such as SITR (Gupta et al., 2025), T3 (Zhao et al., 2024), UniT (Xu et al., 2025), Sparsh (Higuera et al., 2024), and Sparsh-X (Higuera et al., 2025) learn robust, low-dimensional tactile representations without requiring explicit labels. Benchmarks like TacBench (Higuera et al., 2024) evaluate the quality of these embeddings, demonstrating their ability to compress information about contact dynamics, including force fields, slip states, and pose changes, as well as static properties like texture and material. In this work, we use Digit 360 sensors as fingertips for an Allegro Hand mounted on a Franka Panda arm and we use the Sparsh-X model (Higuera et al., 2025) to obtain tactile embeddings, and the Cosmos encoder (Agarwal et al., 2025) to obtain RGB embeddings.

**Action-Conditioned World Models for Real World Robotics:** There has been an influx of training general purpose action-conditioned video-generation models (Hu et al., 2023; Russell et al., 2025; Yang et al., 2024; Bruce et al., 2024; Agarwal et al., 2025; Assran et al., 2025). However, these lines of work focus on show-casing the generation capabilities, and only have limited (if any) results for using these models to control robots.

The majority of previous work on world models applied to real world tasks focuses on visual dynamics models (Agrawal et al., 2016; Byravan et al., 2017; Das et al., 2020; Nagabandi et al., 2020;

Finn et al., 2016; Ebert et al., 2017; 2018; Yen-Chen et al., 2020). Visual dynamics models are either trained directly in pixel-space (Finn et al., 2016; Ebert et al., 2017; 2018; Yen-Chen et al., 2020; Alonso et al., 2024), or in a learned latent space (Watter et al., 2015; Agrawal et al., 2016; Ha & Schmidhuber, 2018; Hafner et al., 2019; Nair et al., 2022; Wu et al., 2023; Tomar et al., 2024; Hu et al., 2024; Lancaster et al., 2024), or in more structured representation spaces such as keypoint representations (Manuelli et al., 2020; Das et al., 2020) or tracked 3D states (Nagabandi et al., 2020). In our work, we train action-conditioned world models in latent states extracted from both RGB and tactile observations.

While several works have investigated learning dynamics model on touch observations (Sutanto et al., 2019; Tian et al., 2019; Ai et al., 2024) there is little work on training world models with vision and touch (Zhang & Demiris, 2023). Furthermore these dynamics models are task-specific - while in our work we aim to train general purpose multi-modal world models that can be used for visual MPC on multiple tasks.

# 3 WORLD MODELS THAT UNDERSTAND CONTACT

Vision-only world models have shown promising capabilities in action steerability and spatial reasoning (Assran et al., 2025; Agarwal et al., 2025), generating plausible rollouts with reasonable robot kinematics and high-quality visuals. These properties make them useful for planning free-space motions. However, simulating object interactions comes with some challenges. Object motion depends on forces invisible to exocentric cameras, and occlusions during grasping, pushing, or placing often cause artifacts such as teleportation, disappearance, or physically implausible dynamics.

We introduce `Visuo-Tactile World Model` (VT-WM), which uses tactile sensing to complement vision to overcome these limitations. Touch provides contact information during occlusion, grounding the model's imagination in contact physics and producing more accurate rollouts for contact-rich manipulation tasks.

## 3.1 WHAT VISION DOESN'T SEE: SENSING CONTACT WITH TOUCH

Touch provides essential local perception, enabling robots to distinguish properties like stiffness, friction, and roughness that are difficult to infer from vision alone. It also captures the dynamics of contact, which is crucial for manipulation tasks. For instance, when manipulating an object in-hand, touch provides context about forces, slip, and subtle pose changes.

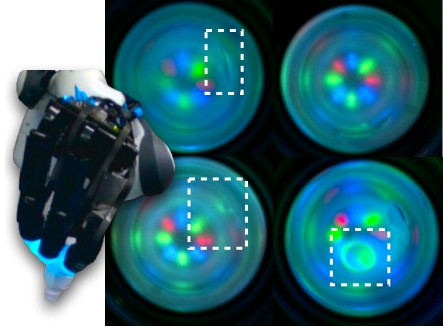

Vision-based tactile sensors typically stream image data at 30-60 FPS, providing rich information about the contact area, including force, shape, and texture features. This tactile information is crucial for disambiguating contact states that from an exocentric camera may appear visually similar. For example, a robot hand's grasp on a cup might look the same from a distance, but tactile sensing can differentiate between a no-contact state,

Figure 2: Tactile images from Digit 360 sensors. White boxes highlight contact while the hand holds a screw.

a subtle touch, or a firm grasp. In this work, we use Digit 360 sensors as fingertips for an Allegro Hand mounted on a robot arm. A visualization of tactile images captured by a Digit 360 sensor is shown in fig. 2.

## 3.2 MULTITASK VISUO-TACTILE WORLD MODEL

### 3.2.1 MODEL ARCHITECTURE

Our visuo-tactile world model is designed to address a key challenge in multimodal robot world models: *how to combine exocentric vision with tactile sensing in order to generate consistent imagined futures*. As shown in fig. 3, the architecture consists of three main components: a *vision encoder*, a *tactile encoder*, and an autoregressive *predictor*.

The vision encoder extracts latent states $s_k$ that capture the robot and its environment from exocentric video. The tactile encoder compresses high-frequency contact feedback into a compact state $t_k$ that emphasizes salient physical interactions. These representations are fused with control

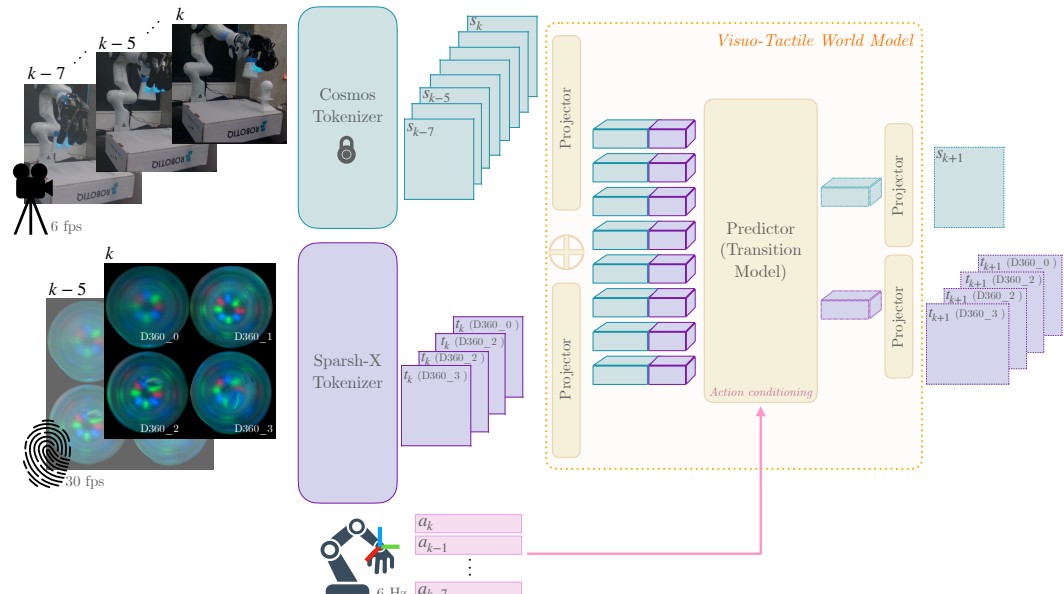

Figure 3: `Visuo-Tactile World Model`. Vision ($s_k$) and tactile ($t_k$) latents, obtained from Cosmos and Sparsh-X encoders, are processed by a transformer predictor given control actions $a_k$ to generate next-step states ($s_{k+1}, t_{k+1}$).

actions and passed to the predictor, a forward dynamics model that estimates the next-step states $(s_{k+1}, t_{k+1}) \sim P_\phi(s_k, t_k \mid a_k)$.

This formulation enables the model to *imagine multiple possible futures under control actions*. Unlike purely visual world models, our predictor leverages tactile signals to disambiguate perceptually identical visual states. For instance, two identical video frames of a robot hand around a cup can lead to different rollouts: if the tactile input indicates contact, the imagined sequence shows the cup being lifted; if not, the cup remains on the table. This tactile-informed disambiguation is significant for planning in contact-rich manipulation.

We frame the predictor as a supervised next-state estimation problem with ground-truth future latents as targets. Both modalities are encoded with pretrained networks: Cosmos tokenizer (Agarwal et al., 2025) for vision and Sparsh-X (Higuera et al., 2025) for Digit 360 tactile sensors. The encoded contexts $s_k$ and $t_k$ are augmented with sinusoidal positional embeddings, and projected into a unified representation $\mathbb{R}^{(b,t,s,d)}$. Vision and tactile tokens are concatenated along the spatial dimension to form a unified input sequence. The predictor then processes these multi-modal tokens through a 12-layer transformer that alternates between two types of attention mechanisms:

**Spatio-Temporal Self-Attention** The model processes vision-touch tokens through factorized attention that operates in two stages: **spatial** attention enables all tokens within a timestep to interact, while **temporal** attention tracks how each token evolves across past timesteps. This factorization efficiently captures both local dynamics and global context while avoiding the $O((THW)^2)$ complexity of full spatiotemporal attention. Action tokens undergo the same factorized attention process.

**Action Conditioning via Cross-Attention** After each self-attention block, vision-touch tokens cross-attend to action tokens to incorporate the robot's control inputs into the predictions. This alternating pattern of self-attention and cross-attention allows the model to iteratively refine its latent states based on both sensory observations and executed actions.

All attention layers employ Rotary Position Embeddings (RoPE) (Su et al., 2023) for relative position encoding. After the transformer, the representations are projected back to their original dimensions through modality-specific output heads, yielding predictions $s_{k+1}$ and $t_{k+1}$.

### 3.2.2 TRAINING VISUO-TACTILE WORLD MODEL

During training, the vision input is a 1.5-second exocentric videoclip (9 frames at 6 fps, $320 \times 192$ resolution), encoded framewise with Cosmos. The tactile input consists of two frames per Digit 360

sensor (four sensors total), covering the most recent 0.16 seconds. This shorter horizon reflects the higher temporal frequency and local nature of contact information, which complements the slower, global context provided by vision. The action input includes changes in proprioceptive state (translation, quaternion rotation) and a binary hand state representing pre-set open/close configurations. We chunk action sequences from 30Hz into groups of 5, with the full chunk of delta-states provided to the predictor. This combination ensures that the predictor models both the external scene and the internal actuation history, a prerequisite for accurate visuo-tactile imagination. The model uses a maximum context length of 9 frames for both vision and touch modalities. Additional details about hyperparameters and training dataset are provided in appendix A.

The model is trained with an objective that combines teacher forcing with autoregressive sampling to balance training stability with long-horizon coherence as proposed by Assran et al. (2025).

**Teacher Forcing Loss.** The primary training signal comes from next-step prediction with ground-truth context. Given a sequence of $T$ frames, we compute:

$$L_{teacher} = \sum_{k=1}^{T-1} ||\hat{s}_{k+1} - s_{k+1}||_1 + ||\hat{t}_{k+1} - t_{k+1}||_1 \qquad (1)$$

where $\hat{s}_{k+1}$ and $\hat{t}_{k+1}$ are predicted from ground-truth states up to time $k$, and $s_{k+1}$ and $t_{k+1}$ are the encoded latents given ground truth observations at time step $k + 1$. This provides dense supervision and stable gradients but can lead to distribution shift during autoregressive rollouts.

**Sampling Loss.** To improve long-horizon generation, we additionally train on sampled trajectories. During training, we sample future states autoregressively for $H$ steps (typically $H = 3 - 5$), then compute predictions conditioned on these sampled states:

$$L_{sampling} = \sum_{k=1}^{H} ||\hat{s}_{k+1}^{sampled} - s_{k+1}||_1 + ||\hat{t}_{k+1}^{sampled} - t_{k+1}||_1 \qquad (2)$$

The sampled states are generated without gradients to prevent training instability. The final loss combines both objectives with equal weighting: $L = L_{teacher} + L_{sampling}$.

### 3.2.3 PLANNING IN IMAGINATION

The action-conditioned nature of our predictor enables the use of the visuo-tactile world model as a simulator within a Cross-Entropy Method (Rubinstein, 1997) (CEM). At each step, the planner samples a population of action sequences $\{a_{k:k+H}^i\}_{i=1}^N$ over a horizon $H$. For each sequence, the predictor autoregressively generates future latents $(s_{k+1:k+H}, t_{k+1:k+H})$. A cost function, defined by energy minimization with respect to a goal image, assigns a score to each trajectory. In practice, this cost can be as simple as an $\ell_2$ distance between the final predicted visual latent $s_{k+H}$ and the latent of the goal image $s_{goal}$. CEM then selects the top-performing fraction of sequences, updates the sampling distribution toward them, and iterates until convergence. The best sequence is then executed on the real robot in an open-loop manner.

We do not provide the tactile modality as a goal signal, thus the planning objective remains purely vision-based. The role of tactile in the VT-WM is to enhance the reliability of the learned world model and, therefore, to improve planning indirectly. First, tactile feedback during training enables the world model to capture contact physics that are difficult to infer from vision alone. Second, when generating rollouts, tactile context in the initial state helps disambiguate visually identical observations (e.g., distinguishing whether the robot is already in contact with an object). This yields more physically consistent imagined futures and more accurate cost evaluations, which we hypothesize translate into higher-quality plans.

## 4 EXPERIMENTS

Our experimental framework evaluates the advantages of visuo-tactile world model (VT-WM) for robot manipulation by addressing the key questions:

- *Contact Perception:* Do VT-WMs better capture object permanence and causal compliance than vision-only WMs, and generate futures consistent with action conditioning?

- *Zero-shot Planning:* Does improved contact perception lead to more reliable zero-shot plan transfer in open-loop execution?
- *Data Efficiency:* Given limited demonstrations, how does fine-tuning a multi-task world model for planning compare to behavioral cloning?

## 4.1 CONTACT PERCEPTION

To evaluate the benefits of incorporating touch, we compare rollouts from a multi-task vision-only world model (V-WM) and our multi-task visuo-tactile world model (VT-WM), conditioned on the same actions and context. The action sequences are drawn from successful demonstrations on the real system, which enables a direct comparison between each model's rollouts and the corresponding ground-truth videos. This setup allows us to assess how well the models capture motion dynamics and the physical plausibility of object interactions. We focus our evaluation on object permanence, causal compliance, and action controllability. These metrics are components of the World

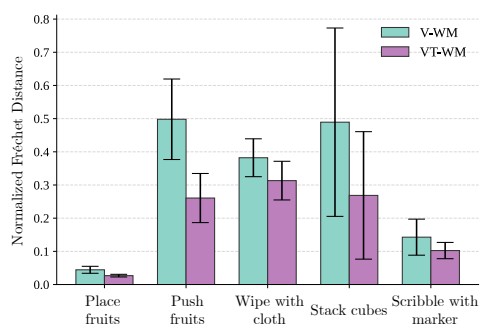

Figure 4: *Object permanence*. VT-WM achieves an average reduction of $\approx 33\%$ relative to V-WM (with 95% CI) of the normalized Fréchet distances for objects in motion.

Consistency Score (Rakheja et al., 2025), a proposed benchmark in the state-of-the-art to evaluate a generative model's ability to maintain coherent and physically plausible futures over time.

**Object Permanence:** This metric assesses a model's ability to maintain a consistent representation of an object's existence and state even when the object is temporarily occluded. As shown in fig. 5, we evaluate whether objects remain represented during heavy occlusion (e.g., during a grasp) and reappear in the correct state once revealed. In the cube-stacking task, VT-WM exhibits stronger object permanence than V-WM: as the blue cube is occluded in the hand during transport and placement, VT-WM preserves its representation, and upon release, the cube re-emerges in the imagined scene at the correct location above the yellow target cube. In appendix B-fig. 13, we visualize the tactile predictions generated by VT-WM, demonstrating the model's ability to maintain congruent visual and tactile representations of object contact.

For quantitative evaluation, we employ CoTracker (Karaev et al., 2024) to track keypoints on the object, providing pixel-level visibility and trajectories. We then compare the normalized Fréchet distance between the ground-truth visual trajectory and the one imagined by V-WM and VT-WM under the ground-truth action conditioning. To ensure comparability, trajectories are expressed relative to the object's initial image position and normalized by the length of the ground-truth trajectory. A lower Fréchet distance indicates that the imagined trajectory more closely reflects the real motion and state of the object, thereby capturing the physical coherence required for object permanence.

Fig. 4 reports normalized Fréchet distances across five tasks. To assess statistical significance, we complement the Fréchet distance analysis with paired *t*-tests across tasks. VT-WM achieves consistently lower distances than V-WM, with statistically significant improvements in *place fruits* ($t = 4.38, p < 0.001$), *push fruits* ($t = 6.06, p < 10^{-6}$), and *cube stacking* ($t = 2.40, p < 0.05$). For *wipe with cloth* and *scribble with marker*, the differences follow the same downward trend but do not reach significance at the 5% level. Quantitatively, VT-WM reduces normalized Fréchet distance by 18–47% across all five tasks, corresponding to an average overall reduction of $\approx 33\%$. These results indicate that VT-WM produces more physically coherent rollouts, with statistically significant gains in tasks requiring reliable object permanence, such as object pushing and stacking.

**Causal Compliance:** This metric evaluates whether changes in object state occur as physically plausible consequences of the robot's actions. A causally compliant model predicts that an object's state changes only when it is subject to external forces. Assessing causal compliance is essential for developing physics-informed world models that respect the principles of contact dynamics and avoid unrealistic motions or deformations.

For a quantitative measure of causal compliance, we use CoTracker to compute the trajectory error of keypoints on objects in the scene that are not subject to any external force and should therefore

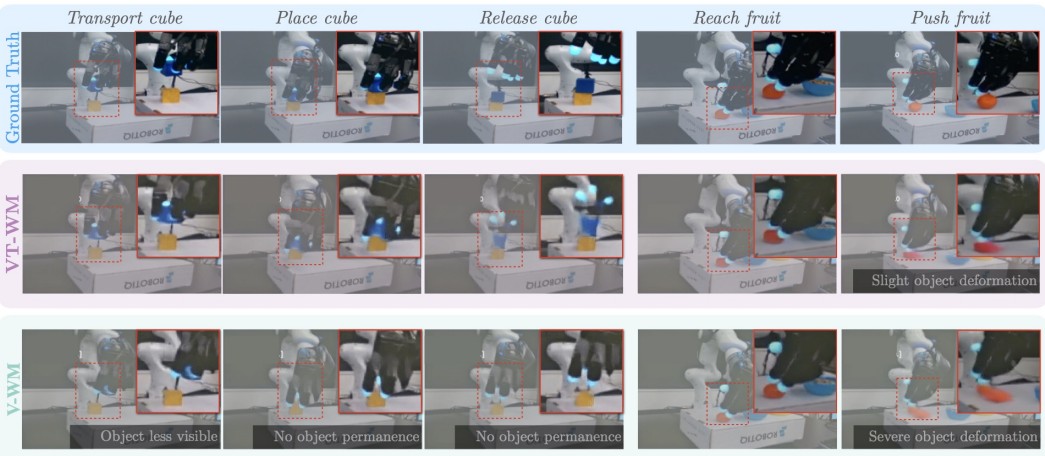

Figure 5: VT-WM preserves object permanence and consistent hand–object interactions during imagination, while V-WM often loses objects or produces severe deformations.

remain stationary. We again use the normalized Fréchet distance between ground-truth and imagined rollouts as our metric. A higher Fréchet distance indicates that the world model hallucinates changes in the position or deformation of these passive objects, thereby violating basic physical laws such as Newton's first law of motion. As shown in fig. 6, VT-WM consistently achieves lower distances than V-WM across most tasks. Paired $t$-tests confirm statistically significant improvements at the 5% level in *place fruits* ($t = 3.66, p < 0.001$), *push fruits* ($t = 2.28, p < 0.05$), and *wipe with cloth* ($t = 2.99, p < 0.01$), while differences in *cube stacking* ($t = 1.75, p = 0.09$), and *scribble with marker* ($t = -1.22, p = 0.23$) are not significant. Quantitatively, VT-WM achieves relative reductions of 43.6%, 16.4%, and 66.1% in *place fruits*, *push fruits*, and *wipe with cloth*, respectively, alongside a smaller improvement in *cube stacking* and a degradation in *scribble with marker*. Overall, VT-WM reduces hallucinated motion by an average of $\approx 29\%$ across tasks, reflecting stronger causal compliance in most scenarios.

In fig. 7 we show snapshots of a trajectory where the robot performs a wiping motion just above a cloth, without making contact. In the ground-truth sequence (top row), keypoints on the cloth remain stationary. In contrast, the V-WM's rollout (bottom row), conditioned on the real actions, shows significant displacement of keypoints and deformations of the cloth. This highlights the V-WM's difficulty to distinguish between contact and non-contact states based on visual input alone. The VT-WM's rollouts (middle row), however, exhibit fewer artifacts and less variation, demonstrating the advantage of tactile sensing in providing the world model with critical physical grounding.

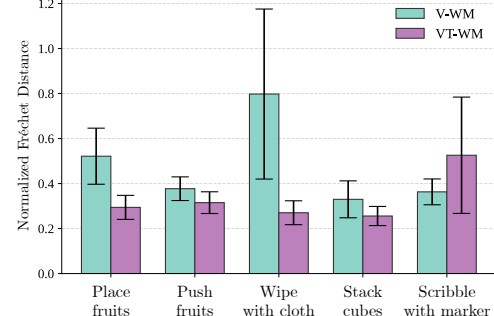

Figure 6: *Causal compliance* evaluates WMs adherence to the laws of motion. Normalized Fréchet distance for static objects (95% CI) shows that VT-WM outperforms V-WM, with an overall improvement of $\approx 29\%$.

In appendix B we showcase the action controllability of the world model. We compare ground-truth trajectories with VT-WM rollouts under the same action sequences and illustrate what the world model *imagines* in terms of contact.

## 4.2 ZERO-SHOT PLANNING TRANSFER TO REAL-WORLD

*Does the improved contact perception of VT-WM translate into superior planning performance on a real robot?*. We hypothesize that VT-WM produces more effective plans for contact-rich tasks. For example, in a cube-stacking task, a physically grounded model should avoid opening its hand while transporting a cube. Similarly, for pushing, the model must recognize that contact is a prerequisite for object motion.

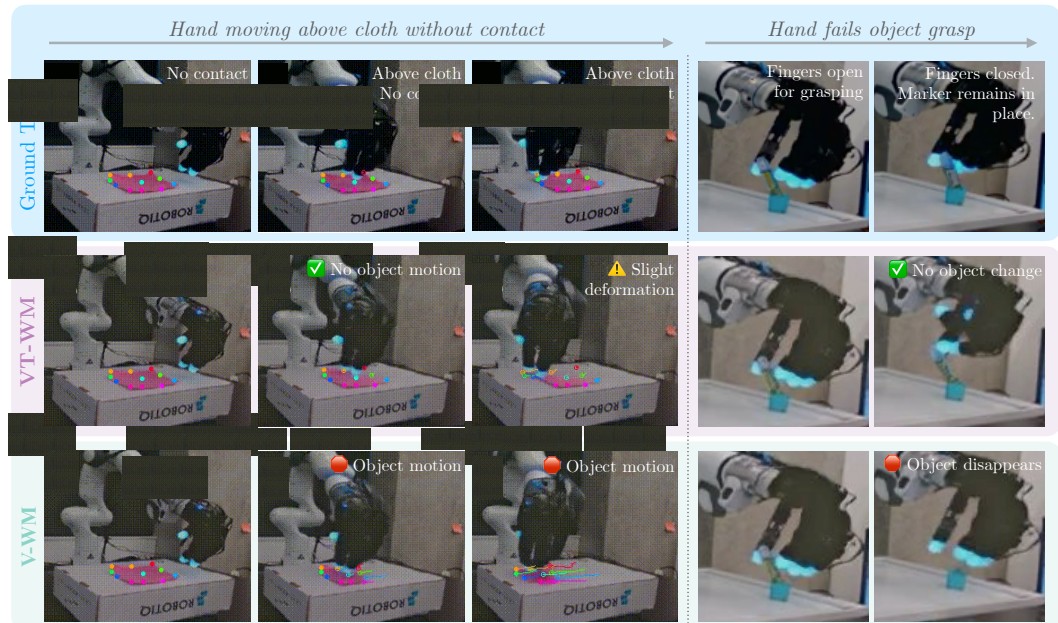

Figure 7: Comparison of rollouts, illustrating that VT-WM prevents spurious motion of objects not subject to forces, whereas V-WM often hallucinates unintended displacements.

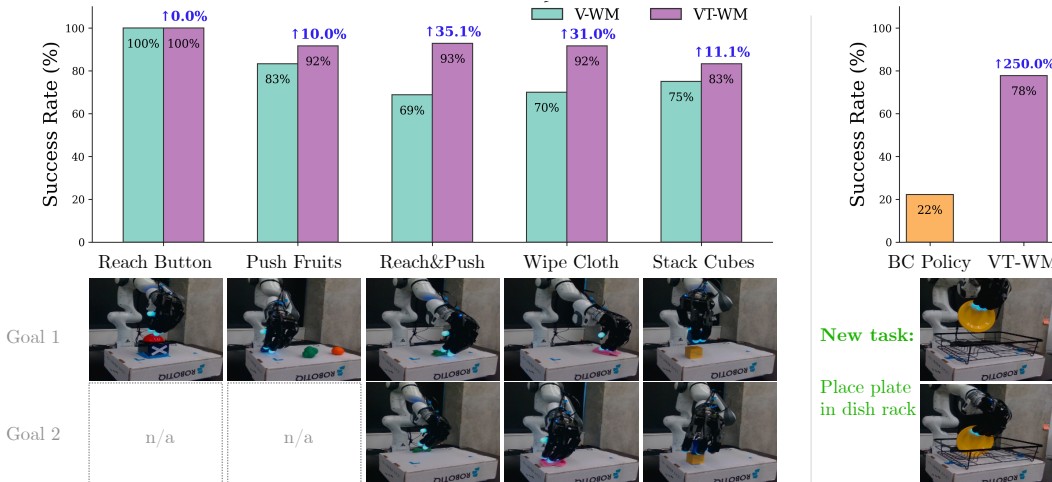

Figure 8: *Left:* Success rate of plans via CEM with VT-WM and V-WM on real robot. For all tasks the VT-WM achieves equal or better performance (blue labels), empirically demonstrating the better planning capability with contact grounding via tactile sensing. *Right:* success rate on new task highlights the data efficiency of VT-WM compared to classical behavioral cloning (BC) policies.

To evaluate this, we employ the Cross-Entropy Method (CEM) (Rubinstein, 1997; De Boer et al., 2005) to solve a goal-conditioned energy minimization problem. The objective is to plan an optimal action sequence over a fixed horizon $H$, where the cost is the distance in latent space between the final predicted visual state $s_{k+H}$ and the goal image latent $s^{goal}$. The search space for CEM is $\mathbb{R}^7$, consisting of 3D translation and 3D orientation of the wrist pose, plus a binary variable for the hand's open/closed configuration.

We evaluate open-loop zero-shot transfer of the generated plans on the real robot. To initialize planning consistently, the initial RGB and tactile embeddings are passed as context to the world model, indicating whether optimization should begin from an in-contact or no-contact state. Both VT-WM and V-WM are tested on five tasks of increasing difficulty: *reach button*, *push fruits*, *reach & push*, *wipe cloth*, and *stack cubes*. The first two are single-goal tasks, while the latter three involve multiple subgoals.

Fig. 8(left) reports success rates, averaged over five trials per task from distinct initial conditions. The results confirm the superior planning capability of VT-WM across all tasks, supporting our hypothesis that a contact-aware model generates more effective plans. On simple tasks such as *reach button*, both models achieve 100% success, consistent with prior visual world models for robot manipulation such as V-JEPA-2AC (Assran et al., 2025). However, the benefits of tactile input become increasingly evident in contact-rich tasks: VT-WM improves success rates by 10% on *push fruits*, 35% on *reach & push*, 31% on *wipe cloth*, and 11% on *stack cubes*. These gains are most pronounced in multi-step tasks involving sustained contact, where vision alone is insufficient to inform about the object state during planning. In appendix C, we describe the experimental setup for each task and present a qualitative evaluation of the planned trajectories and their corresponding real-world executions.

### 4.3 DATA EFFICIENCY

For a new task with a limited number of successful demonstrations, *how does fine-tuning a multi-task world model for planning compare to training a task-specific behavioral cloning (BC) policy?* We hypothesize that VT-WM can extract task structure even in low-data regimes, since it already encodes contact dynamics from prior tasks. For example, insertion may involve a new object, but concepts such as alignment and adjusting contact with a receptacle can be reused. In contrast, a BC policy must learn both spatial and contact reasoning from scratch.

We collect 20 demonstrations of the task "place plate in the dish rack," which requires transporting the plate and inserting it between the racks. We augment our multi-task dataset (see appendix A.0.1) with the new sequences and continue training VT-WM, while also training a task-specific BC policy ACT (Zhao et al., 2023) that outputs action chunks over a fixed horizon. For VT-WM, we use CEM planning and zero-shot transfer to the real robot. The task is divided into two subgoals: alignment and insertion (see fig. 8). The BC policy is deployed in closed loop, where at each timestep it receives the latest RGB and tactile inputs and executes the first action of the predicted chunk.

We evaluate both methods in nine real-robot trials, randomizing the initial plate-in-grasp pose. As shown in fig. 8(right), VT-WM planning achieves a 77% success rate, compared to 22% with BC. These results highlight the data efficiency of multi-task world models and their advantage over task-specific policies. Moreover, failure modes differ, for instance, VT-WM mostly places the plate beside the rack, whereas 57% of BC failures involve the robot never reaching the rack at all.

## 5 DISCUSSION AND CONCLUSION

Visuo-Tactile World Model (VT-WM) leverages tactile sensing to complement vision, enabling world models for robot manipulation to be grounded in contact. While vision-only world models (V-WMs) have shown promise in spatial reasoning and capturing robot kinematics, they are prone to hallucinate object interactions, with failures such as object disappearance, teleportation, or unrealistic deformations. By integrating fingertip tactile sensing with exocentric vision, VT-WM grounds imagination in the physics of contact, producing more accurate rollouts and capturing core concepts such as object permanence under occlusion and compliance with physical laws.

We studied the gains of multimodality in VT-WM over V-WM through three key questions. First, *does adding touch improve a world model's understanding of contact?* Comparing autoregressive rollouts under identical action sequences, we found that VT-WM preserved object permanence, even when objects were occluded by the robot hand, and maintained resting states for objects not subject to external forces. Quantitatively, VT-WM reduced normalized Fréchet distance by 33% on average relative to V-WM, better reflecting true object dynamics.

Second, *does contact grounding improve planning?* Using CEM-based imagination for goal-conditioned planning, we zero-shot transferred plans to a real robot under randomized initial conditions. While both models performed similarly on free-space tasks such as reaching, VT-WM achieved up to 35% higher success rates in contact-rich tasks requiring precise hand–object interaction, including pushing, wiping, and stacking.

Third, *is VT-WM data efficient when targeting a new task?* To compare with task-specific behavioral cloning (BC), we fine-tuned VT-WM on just 20 demonstrations of a plate-insertion task. VT-WM reached a 77% success rate, over $3\times$ higher than BC, by reusing priors from previously learned contact-rich tasks such as alignment and insertion. This highlights its ability to efficiently adapt to new tasks with limited data.

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

APPENDIX

# A  TRAINING VISUO-TACTILE WORLD MODEL

### A.0.1  TRAINING DATASET

To train our multi-task visuo-tactile world model, we collect a dataset of teleoperated robot arm trajectories performing fundamental contact-rich manipulation actions, such as pick and place, pushing, and insertion. Our hardware setup consists of a table-top Franka Panda arm with an Allegro Hand as the end-effector and a Digit 360 sensor mounted on each fingertip. An exocentric view from a camera captures the global context of the robot's interaction with objects on the table.

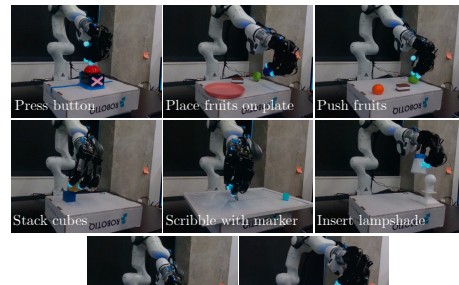

Through teleoperation, we collect a diverse set of trajectories, without discriminating between successes and failures, for eight distinct contact-rich tasks (see Fig. 9): pick and place on a plate, reach and press a button, push, wipe with a cloth, lampshade insertion,

Figure 9: Multitask Vision-Tactile Dataset. Trajectories for training the world model collected via teleoperation, including both successful and failure sequences.

table leg insertion, cube stacking, and scribbling with a marker. For each task, we recorded successful and failure demonstrations. Each sequence contains multimodal data streams: proprioceptive information (wrist pose, joint positions), exocentric video from the camera, and video from each Digit 360 fingertip sensor. All data streams were synchronized using timestamps and downsampled to 6 FPS for training the world model. Our training dataset for V-WM and VT-WM consists of 124 demonstrations totaling 112k datapoints, with each demonstration averaging 40 seconds. For validation, we use 26 demonstrations spanning all tasks, comprising 17k datapoints.

## A.1  TRAINING PARAMETERS

The model is optimized using AdamW (Loshchilov & Hutter, 2019) with parameters $\beta_1 = 0.9$, $\beta_2 = 0.95$ and a weight decay of 0.01. We use a learning rate scheduler with linear warmup for the first 10,000 gradient updates to a peak learning rate of $3e-4$, followed by cosine decay to $3e-7$ over a total of 80,000 updates. We use an effective batch size of 64 distributed over 32 A100 GPUs. We found that fine-tuning the Sparsh-X encoder was beneficial to account for sensor-specific variations, such as those arising from manufacturing tolerances and elastomer wear, while the Cosmos Tokenizer (Agarwal et al., 2025) was kept frozen during training. Our visuo-tactile world model has a total of 173M parameters, of which 96M were trained.

# B  CONTACT PERCEPTION WITH VISUO-TACTILE WORLD MODEL

We corroborate the world model's capacity to generate future states that are a reliable and predictable consequence of the given action conditioning. We study action controllability qualitatively by visualizing rollouts under simple, disentangled action commands: moving the end-effector along the Cartesian axes ($\pm x$, $\pm y$, $\pm z$) and opening/closing the hand. Actions conditioning is given to the VT-WM as deltas in the robot's proprioceptive state.

We observe in Fig. 10 that the VT-WM produces coherent rollouts aligned with the commanded actions. Translations along each axis result in consistent directional displacements of the end-effector in imagination (notice the reference frame in the figure), while hand open/close commands lead to corresponding changes in finger configurations. Notably, these behaviors emerge from the learned dynamics rather than explicit supervision of axis-aligned motion, indicating that the model internalizes the action-conditioned structure of the robot's kinematics. We compare ground-truth trajectories with VT-WM rollouts under the same action sequences and illustrate what the world model *imagines* in terms of contact.

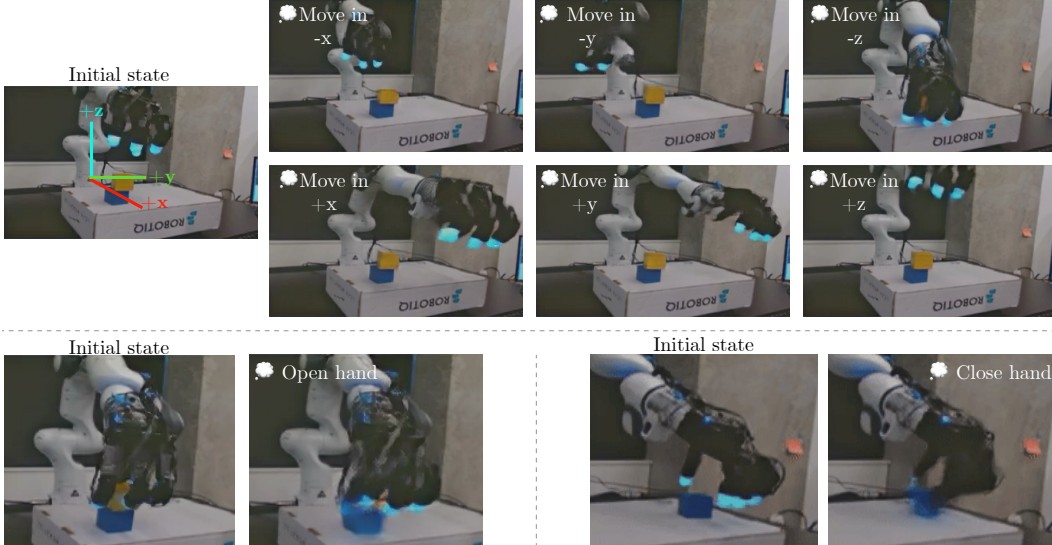

Figure 10: Visuo-Tactile World Model generates rollouts aligned with commanded actions along reference axes ($\pm x$, $\pm y$, $\pm z$) and for hand open/close.

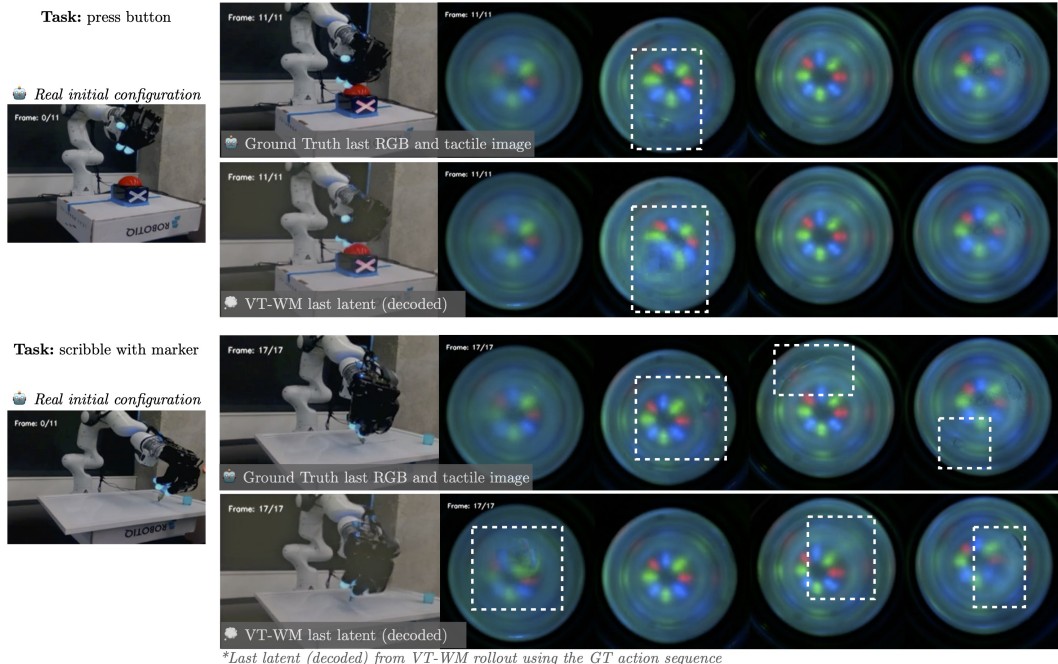

Figure 11: Visuo-Tactile World Model rollouts conditioned on ground-truth action sequences. Predicted visual states closely match the final RGB observations, while predicted tactile states capture plausible contact events and finger–object interactions.

To illustrate the predictive capability of the visuo-tactile world model, we evaluate rollouts conditioned on real robot action sequences. Specifically, we use held-out demonstrations from two tasks in our dataset: *press button* and *scribble with marker*. For each task, the VT-WM is queried autoregressively using the ground-truth sequence of control deltas.

Fig. 12 compares snapshots of a `Visuo-Tactile World Model` rollout for the *insert table leg task, when grasping the object.* Fig. 11 compares the final predicted states with the corresponding real-world outcomes. Since the model produces latent representations of future visual and tactile observations, we employ pretrained decoders to reconstruct these latents for visualization. Across both

tasks, the predicted visual states closely resemble the final RGB images of the real trajectories. The predicted tactile states also capture the key interaction events: although slight differences appear in the precise location of per-finger contacts, the rollouts consistently indicate whether contact occurs and depict plausible patterns of hand–object interaction. This demonstrates that the VT-WM, when guided by real action sequences, generates in imagination physically meaningful futures across both visual and tactile modalities.

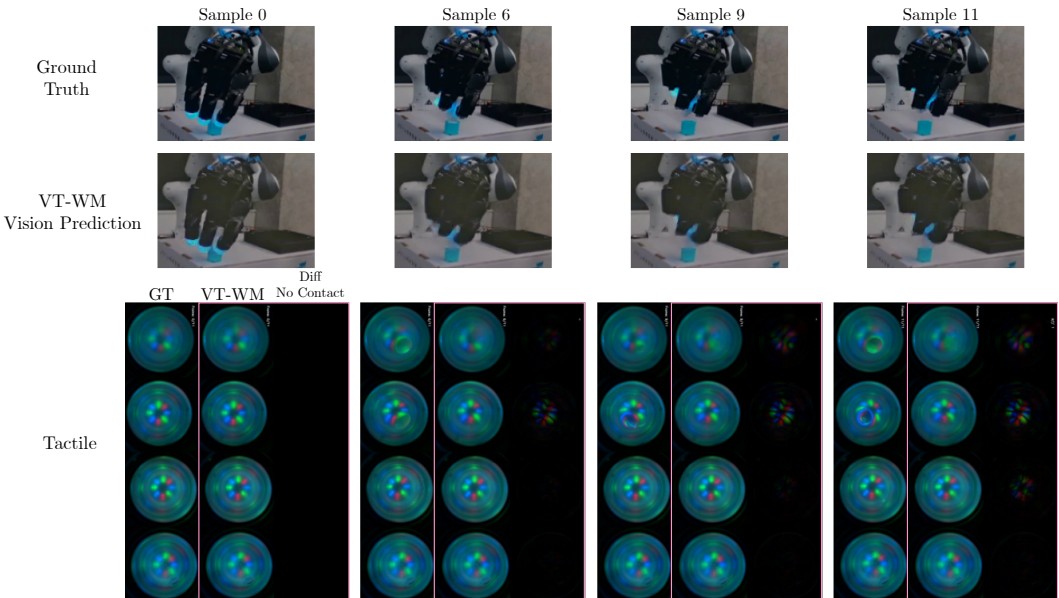

Figure 12: Snapshots of a Visuo-Tactile World Model rollout for *insert table leg* task conditioned on ground-truth action sequences for a 2s horizon. *Top:* ground truth vision state. *Middle:* predicted vision state across rollout. *Bottom:* ground truth tactile signatures, predicted tactile and its difference with respect the no contact state. Notice that fingers that are in contact match between ground truth and VT-WM predicted signatures.

By producing consistent visuo-tactile rollouts under real control sequences, VT-WM demonstrates the ability to represent both global visual context and local contact dynamics in a unified predictive framework useful for planning.

## C   ZERO-SHOT PLANNING WITH WORLD MODELS

### C.1   CEM ALGORITHM FOR PLANNING WITH WORLD MODELS

The algorithm 1 performs planning in a world model (WM) imagination using the Cross-Entropy Method (CEM). Given a goal image and the current multimodal context (vision and tactile), the algorithm first encodes these inputs into latent representations. CEM is then used to optimize a sequence of actions over a finite prediction horizon by iteratively sampling action sequences (particles), rolling them out in the world model, and evaluating their predicted visual outcomes against the goal latent state using an $\ell_2$ distance. The top-performing action sequences are used to update the mean and variance of the action distribution, refining the search over multiple iterations. After convergence, the best action sequence is executed on the robot, and the process can be repeated over several trials with updated context.

Next, we describe the goal of each of the tasks we use to evaluate the planning capabilities of the world models and discuss their results.

**Reach Button:**   In this task, the robot must approach and press the center of a button starting from varied initial poses directly above it. Success therefore requires planning a sequence of actions that align the end-effector laterally with the button and then move downward to establish contact. Fig. 14 illustrated the plans produced by each world model using CEM rollouts in imagination,

---

**Algorithm 1** Planning in WM imagination via CEM

---

**Require:** WM (world model)
**Require:** Goal Image $X_{rgb}^{goal}$
**Require:** Context (current state) $X_{rgb}^0$, $X_{touch}^0$
   $f \leftarrow 6$                                                                ▷ WM frequency
   $H \leftarrow 2$                               ▷ Prediction horizon in seconds
   $P \leftarrow 36$                   ▷ Number of particles for CEM algorithm
   $N \leftarrow 10$                   ▷ Number of iterations for CEM algorithm
   $d \leftarrow 7$            ▷ Action dimensionality [X,Y,Z,roll,pitch,yaw,gripper]
   max-trials $\leftarrow 3$                 ▷ Number of calls to CEM algorithm
   **for** trials $<$ max-trials **do**
       Update context (current state) $X_{rgb}^0$, $X_{touch}^0$          ▷ Read from sensors
       $Z^{goal} \leftarrow$ vision-encoder$(X_{rgb}^{goal})$          ▷ Encode goal image
       $Z_{rgb}^0 \leftarrow$ vision-encoder$(X_{rgb}^0)$ and $Z_{touch}^0 \leftarrow$ touch-encoder$(X_{touch}^0)$     ▷ Encode context
                                         ▷ Initialize CEM action distribution parameters
       $\mu \leftarrow zeros(1, H * f, d)$ and $\sigma \leftarrow ones(1, H * f, d)$
       best-cost $\leftarrow \infty$
       best-action $\leftarrow None$
       **for** $n < N$ **do**
                                           ▷ Generate action particles
          $actions \leftarrow (\mu.repeat(N) + \sigma.repeat(N)) * rand(N, H * f, d)$
          $\hat{Z}_{rgb}^{1:H}, \hat{Z}_{touch}^{1:H} \leftarrow$ WM.rollout$(Z_{rgb}^0, Z_{touch}^0, actions)$       ▷ Rollout WM
          $costs \leftarrow \ell_2 \left( Z^{goal}, \hat{Z}_{rgb}^H \right)$       ▷ Compute distance with target latent state
          elite-actions $\leftarrow actions[topk(costs)]$       ▷ Choose top 5 particles with lowest cost
                                            ▷ Update distribution parameters
          $\mu \leftarrow$ elite-actions.$mean()$ and $\sigma \leftarrow$ elite-actions.$std()$
          **if** $costs.min() <$ best-cost **then**
              best-action $\leftarrow actions[costs.argmin()]$ ▷ Found a new action that gets closer to goal
          **end if**
       **end for**
       Execute sequence of robot commands from best-action
   **end for**

---

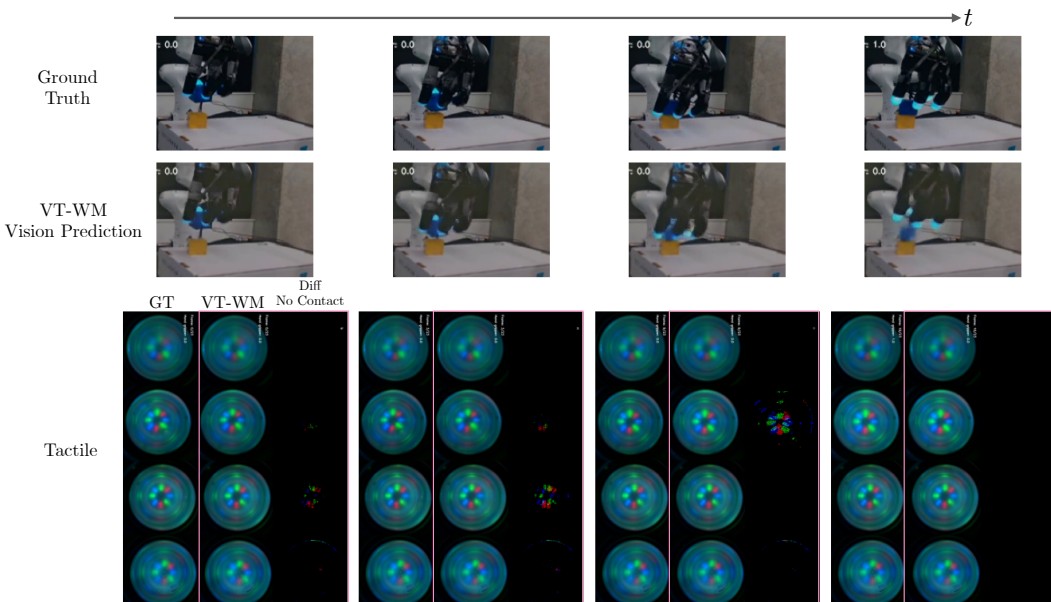

Figure 13: Snapshots of a Visuo-Tactile World Model rollout for *cube stacking* task. *Top:* ground truth vision state. *Middle:* predicted vision state across rollout. *Bottom:* ground truth tactile signatures, predicted tactile and its difference with respect the no contact state. Notice that fingers that are in contact match between ground truth and VT-WM predicted signatures.

alongside the corresponding executions on the real robot. We observe that both V-WM and VT-WM generate feasible trajectories that transfer zero-shot to the real system. This is expected since reaching primarily involves spatial reasoning and gross kinematic alignment, which vision alone can capture reliably.

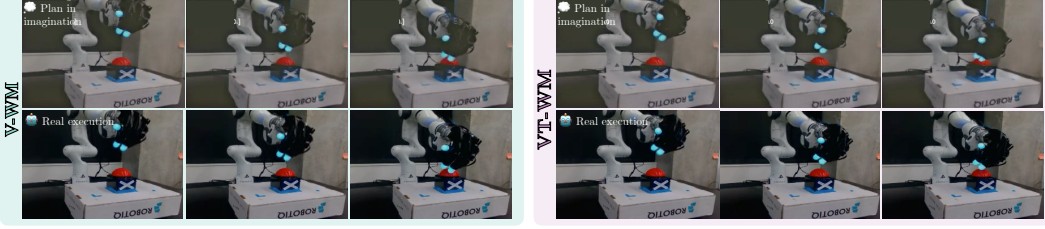

Figure 14: Reach Button task. Plans generated by V-WM and VT-WM with CEM in imagination (top) and their zero-shot executions on the real robot (bottom). Both models produce feasible trajectories, as reaching relies mainly on spatial reasoning and kinematic alignment.

**Push Fruits:** In this task, the robot hand begins directly in front of a target object that must be pushed downwards (toward the robot base). A successful plan requires maintaining persistent but gentle contact, allowing the object to slide across the table rather than topple.

Fig. 15 compares imagined rollouts and real executions for both models. While both V-WM and VT-WM produce plausible plans, we observe notable artifacts in the imagined rollouts, most prominently visual distortions of the green fruit when the hand occludes it. These artifacts are less pronounced in VT-WM, which better preserves the object's geometry in imagination. The deployment of the V-WM plan not only results in shorter object displacement but also lead to physical failures in execution, where the object occasionally topples instead of sliding without orientation changes.

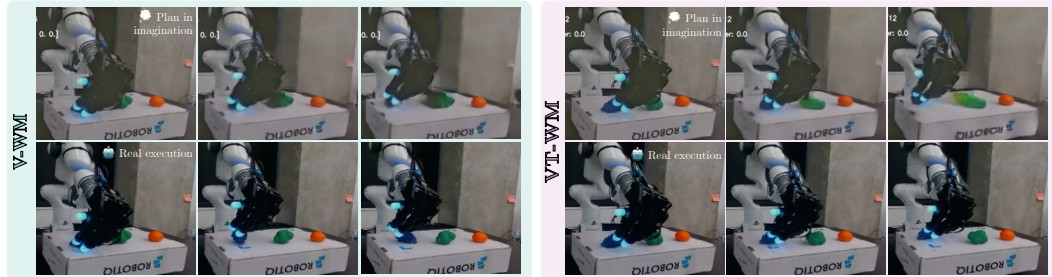

Figure 15: Push Fruits task. VT-WM preserves object geometry in imagination and transfers to stable sliding, while V-WM introduces distortions and often causes toppling in execution.

**Reach & Push:** This task requires a two-stage plan, first reaching the object to establish contact, then pushing it downward toward the robot base. Both subgoals are illustrated in Fig. 16, which shows the imagined plans and real executions.

In the V-WM rollout, the hand consistently hovers slightly above the object during the reach phase. As a result, the subsequent push proceeds without contact, and the execution on the real robot fails to move the object. By contrast, the VT-WM rollout explicitly brings the hand into contact during the reach, enabling the push plan to apply move the object effectively. When deployed, this produces the desired behavior, with both the reach and push subgoals successfully achieved. This highlights how tactile grounding resolves cases of visual aliasing, ensuring reliable contact in imagination and execution.

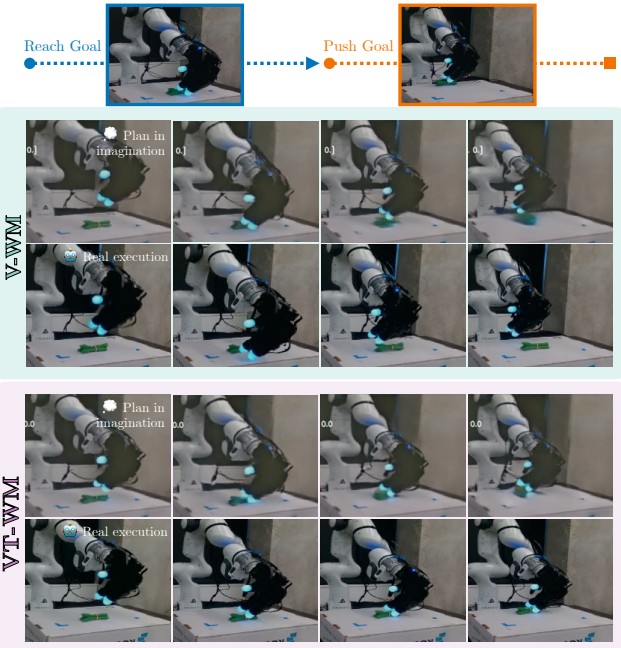

Figure 16: Reach & Push task. V-WM fails to establish contact, leading to ineffective pushes, while VT-WM ensures contact in imagination and execution, successfully completing both subgoals.

**Wipe Cloth:** This task consists of two subgoals, first reaching the cloth to establish contact, and then wiping it horizontally across the table. Both stages are illustrated in Fig. 17, which compares imagined rollouts with real executions.

In the V-WM rollout, the reach phase frequently results in the hand hovering slightly above the cloth, leading to ineffective wiping during execution. Even when a wiping trajectory is imagined, the visualizations exhibit noticeable artifacts such as geometric distortions of the cloth and hand. These artifacts reflect the model's uncertainty about contact dynamics and correspond to execution failures where the cloth barely moves.

In contrast, the VT-WM rollout shows clearer geometry and maintains consistent contact with the cloth in imagination. As a result, the subsequent wiping action produces a stable horizontal displacement of the cloth when deployed on the real robot. This example underscores the advantages of visuo-tactile world model in tasks that require sustained contact to manipulate objects.

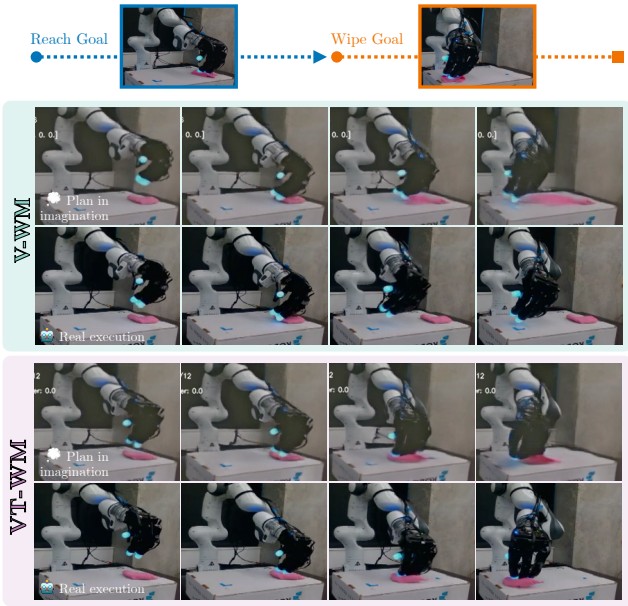

Figure 17: Wipe Cloth task. V-WM rollouts show artifacts and miss contact, leading to ineffective wiping, while VT-WM maintains contact and produces consistent cloth displacement in execution.

**Stack Cubes:** This task requires transporting a blue cube to the stacking location and then placing it stably on top of a yellow cube. Both subgoals are illustrated in Fig. 18, which shows imagined rollouts alongside real executions.

While the V-WM generates reasonable transport trajectories, failures arise during placement. In imagination, the cube intermittently disappears from the hand, revealing artifacts that indicate the model is tracking only the hand–scene geometry (e.g., alignment with the target yellow cube) rather than maintaining a consistent hand–object relationship. This disconnect leads to execution failures, where the cube is not reliably placed.

However, visuo-tactile world model accurately captures the object–hand interaction, throughout both transport and placement, the cube remains consistently represented in the rollout. When transferred zero-shot to the real robot, these plans result in stable stacking, highlighting the advantage of VT-WM for tasks that demand precise, contact-rich manipulation.

# D    LIMITATIONS

While our results demonstrate clear benefits of visuo-tactile world model, following limitations point to promising directions for future research. First, our tactile modality is limited to vision-based tactile sensing, specifically the Digit 360 sensor. However, the VT-WM framework applies to other tactile modalities as well, provided that an appropriate pretrained tactile encoder is available. Second, evaluation of contact perception uses unseen robot trajectories but only within tasks from the training distribution, leaving open the question of how well the model generalizes to entirely novel

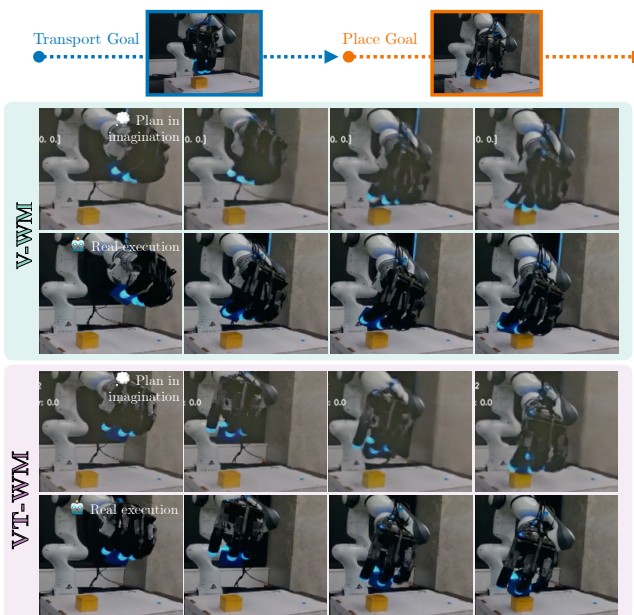

Figure 18: Stack Cubes task. V-WM imagined rollouts during planning lose track of the cube for the placement subgoal, leading to failed stacking, while VT-WM preserves hand–object interaction and transfers to successful stacks.

manipulation tasks or object characteristics. Third, our planning experiments randomize initial robot states but are limited to the same scene and objects, without testing generalization to objects with different visual or physical properties such as size, shape, or color. Planning with world models via CEM remains computationally expensive, as it requires generating many autoregressive rollouts per particle. This leads to open-loop execution in trajectory chunks, unlike classical policies that operate in closed-loop at higher control frequencies. Finally, our comparison against a single task behavior cloning (BC) policy does not fully rule out the possibility that a multi-task BC policy could also exhibit strong data efficiency for the new task.

# E ADDITIONAL NOTES

**About the use of large language models:** Large Language Models (LLMs) were used exclusively to assist with grammar correction and refinement of writing style (flow, academic tone, and conciseness), based on drafts authored by the researchers. LLMs were not employed for data generation, or in any stage of the proposed model's design, training, or evaluation.

