# OpenReview forum: "Visuo-Tactile World Models"
_ICLR.cc/2026/Conference — Submitted to ICLR 2026_

### Official Review · Reviewer_8r8W · 2025-10-24

**Soundness:** 3
**Presentation:** 3
**Contribution:** 3
**Rating:** 6
**Confidence:** 4

**Summary:**

This paper proposes Visuo-Tactile World Model (VT-WM) for contact-rich robot manipulation. The model encodes RGB images and images from fingertip vision-based tactile sensors, performs action-conditioned latent rollouts with a transformer, and plans via CEM toward a vision-only goal image.

**Strengths:**

- paper is well structured and easy to read.
- motivation is good. Vision only world models do lack the capability to handle contact-rich scenarios
- The examples are compelling and convincing, eg. the robot arms grasping blue cube.
- The experiment design and results are interesting.

**Weaknesses:**

- The paper presents a visuo-tactile world model, but the tactile stream is limited to vision-based tactile images (Digit 360). That is a narrow slice of touch compared to other modalities (force/torque, capacitive arrays, pressure taxels, vibration, thermal, shear/strain, proprioception at contact, etc.). As written, the title/claims risk overgeneralizing beyond the evaluated sensing class.

- Most data come from similar tabletop setups and Digit 360 configuration. This couples performance to a specific sensor, mount, and fingertip geometry, which can limit transfer. See prior work (e.g., AnySkin, Bhirangi et al., ICRA 2025; see Fig. 7) for the cross-sensor gaps for vision-based tactile.

**Questions:**

- Could you clarify the taxonomy (e.g., “vision-based tactile”) in title/abstract or limitations; discuss what would change for non-vision tactile sources (signal statistics, encoder design, synchronization, calibration)?
- The rollout method is quite compute-heavy, how long does it take to finish a simple task?
- How does the performance of VT-WM compare with VT diffusion policy baseline?

---

> ### Author Response · Authors · 2025-11-21
> **Response to Reviewer 8r8W**
>
> We thank the reviewer for the thoughtful and constructive feedback, and for highlighting the clarity of the paper, strong motivation, and compelling experimental results. We address the raised concerns point-by-point below.  We have also updated the manuscript based on your suggestions.
>
> ### Vision-based Tactile Modality Scope and Generalization other touch modalities (Weakness 1, Weakness 2, Question 1)
>
> We completely agree with your assessment regarding the scope of the tactile modality that we use in the paper.
>
> **Taxonomy Clarification:** We have updated the abstract to reflect that our experiments and claims hold for vision-based tactile sensors, in particular Digit 360. We also added this as a limitation of our work (see Appendix D), since there are a plethora of other touch sensing modalities that can be used based on the hardware interface and task.
>
> **Sensor Diversity:** The core contribution of our work is to demonstrate the value of the touch modality for contact-grounded world model imagination. We recognize that the lack of a standardized tactile sensor makes cross-hardware generalization a persistent challenge in the field. For this work we used vision-based tactile sensors, since pre-trained encoders are already available ([Sparsh-X](https://akashsharma02.github.io/sparsh-x-ssl/))  and authors have prior experience using gelsight-like sensors.
>
> **Adaptation to Other Touch Modalities (non-vision tactile source):** As you suggest, the sensor type dictates the required encoder architecture.
>
> For other vision-based tactile sensors (e.g., GelSight), the current vision transformer-based encoder is appropriate, but retraining or fine-tuning would be necessary. Also, pre-train encoders are already available and can be swap out, such as [T3](https://github.com/alanzjl/t3), [AnyTouch](https://github.com/GeWu-Lab/AnyTouch) and [Sparsh](https://github.com/facebookresearch/sparsh).
>
> For non-vision tactile sensors (e.g., force/torque, capacitive arrays), the encoder must change to process different signal statistics (e.g., time-series data). For magnetic-skin-based sensors, pre-train encoders can also be used (e.g. [Sparsh-Skin](https://github.com/facebookresearch/sparsh-multisensory-touch)). However, the forward dynamics model architecture of VT-WM (the transformer that performs action-conditioned latent rollouts) remains sound and only requires training on the new latent space provided by the corresponding touch encoder.
>
> **Data Diversity:** We acknowledge that our dataset is constrained by our single hardware setup. Expanding the dataset to incorporate touch diversity across multiple sensors and environments is a critical and exciting direction for future research.
>
>
> ### The rollout method is quite compute-heavy, how long does it take to finish a simple task?
>
> This is an important point, and we agree that computational expense is the main bottleneck for World Model-based planning in robotics.
>
> *Quantification:* Our current planning setup, which uses CEM with 36 particles for 10 iterations, takes approximately 66 seconds per planning call. The total execution time for a task depends on the number of subgoals and re-trials.
>
> *Limitation Acknowledgment:* As a result, the execution is not currently real-time. We have formally acknowledged this limitation in our original submission in Appendix D (Limitations) and emphasize that improving the computational efficiency of planning with world models is an active and critical line of research for future work in this field.
>
> ### How does the performance of VT-WM compare with VT diffusion policy baseline?
>
> Our primary experimental focus was to validate the data-efficiency enabled by the VT-WM's learned prior dynamics, in comparison to a specialist policy that lacks this prior. We chose Action Chunking with Transformers (ACT), a strong and representative Behavioral Cloning (BC) baseline, for its simplicity and effectiveness in tabletop manipulation tasks.
>
> *Comparison Rationale:* While a Diffusion Policy (DP) is a compelling alternative, the performance gap between modern BC methods (like ACT) and DP is often task and data dependent. We believe our comparison against a strong, lightweight BC baseline is sufficient to highlight the key benefit of our approach.
>
> *Key Result:* The most relevant finding is that for a new task (plate-insertion), fine-tuning VT-WM on only 20 demonstrations resulted in a 77% success rate, outperforming the specialist BC policy by over 3.5x. This result strongly demonstrates that the VT-WM's contact-grounded imagination provides a highly data-efficient mechanism for planning on new, contact-rich tasks.

---

### Official Review · Reviewer_KZLz · 2025-10-30

**Soundness:** 3
**Presentation:** 3
**Contribution:** 3
**Rating:** 8
**Confidence:** 4

**Summary:**

The paper proposes training world models with tactile sensing to enable world models that remain grounded in physical contact. The paper validates the superiority of tactile world model in contact-rich scenarios and also shows the usefulness of such world models for real world planning through CEM.

**Strengths:**

- The paper tackles an important problem of developing physically grounded world models using tactile sensing as an additional signal.
- The experiments in the paper are nicely organized with an effort to assess statistical significance of the results.
- The authors also show the ability to perform zero-shot planning using their proposed world model using CEM.
- Further, the authors highlight the data efficiency of training models models for real world planning as opposed to specialized task-specific policies.

**Weaknesses:**

Including both weaknesses as well as questions tied to the weaknesses below.
- In Section 4.3, what if one trained a multitask BC policy instead of a task-specific policy. I am assuming  the authors have actions available from the training data collected for training the world model. I am curious to see if multitask BC training exhibits similar data efficiency.
- I would be curious to see the difference in performance if the world model that takes tactile readings as input but doesn’t predict tactile outputs. This will throw some light on the need for reconstructing all observation modalities versus only reconstructing vision (with all modalities as input).

**Questions:**

It would be great if the authors could address questions in the weaknesses section.

---

> ### Author Response · Authors · 2025-11-21
> **Response to Reviewer KZLz**
>
> Thank you for your very positive and encouraging review, particularly noting the importance of the problem, the organization of our experiments, the statistical rigor, and the demonstrated zero-shot planning capability. Your comments are insightful, and we appreciate your rating of *accept, good paper.*
>
> Below are the responses to address your two questions regarding the data-efficiency BC baseline and the necessity of tactile output prediction.
>
>
> ### Data Efficiency and Multitask BC Baseline
>
> This is a valid and excellent point regarding the baseline comparison. We acknowledge that comparing against a single task BC policy does not fully rule out the possibility that a multi-task BC policy could also exhibit strong data efficiency for the new task. We included this in our Limitations section (Appendix D).
>
> We specifically chose to compare against a specialist policy (ACT trained for a single task) because, in practice, this represents the strongest possible non-pretrained baseline for that specific task. The VT-WM approach is generative, learning a predictive model of the environment. Even a strong multi-task BC policy is fundamentally discriminative, since it learns state-to-action mapping. The WM's inherent ability to simulate counterfactual scenarios (what happens if taking action $A$ vs. action $B$?) is presumably what enables better data efficiency via its use in the CEM planning loop. This capability is absent in a standard BC policy, regardless of whether it is trained on one task or multiple tasks.
> &nbsp;
>
> ### Necessity of Tactile Output Prediction
>
> This is an interesting observation. Our motivation began with the idea of a world model that can imagine how something would feel, much like how we can precisely sense the texture of a tennis racket’s grip and notice whether it feels too slick or too tacky.
>
> Given our VT-WM implementation, for planning the WM needs to generate an autoregressive rollout (predicting a sequence of future states). The prediction at time $t$ becomes part of the context for the prediction at time $t+1$. Since the input state at every step requires both vision and tactile latents (as context for the forward model), the VT-WT must predict the next tactile state to continue the rollout. If we only predicted the vision output, the WM would lose the necessary tactile information to predict the next step's state, breaking the contact-grounding in the planning process.

---

> > ### Comment · Reviewer_KZLz · 2025-11-27
> >
> > I thank the authors for the rebuttal. I would like to keep my rating.

---

### Official Review · Reviewer_MMZP · 2025-10-31

**Soundness:** 3
**Presentation:** 3
**Contribution:** 2
**Rating:** 4
**Confidence:** 5

**Summary:**

The paper introduce a visuo-tactile world models which help capture physics of contact through touch. The authors motivation based on the leverage tactile sensing to complement the vision. The authors aim to train a general purpose multi-modal world models for planning. The tactile is not in the final planning goal but in a more implicit way to help the planning.

While the zero-shot planning is appreciate, the paper failed to show the benefit of using visuo-tactile world model over BC policy training from scratch. The paper do not fully support the "data efficiency" due to baseline selection(baseline train from scratch). Also, the paper do not fully justify the usage of tactile information due to task selection(tasks are simple which could be completed with vision only policy).

**Strengths:**

The paper is well presented, with clear illustrations and a coherent narrative. It is easy to read and effectively conveys the authors’ key ideas. The results are clearly reported.

**Weaknesses:**

Minor weakness:

1. Additional visualization of the tactile signals would help readers grasp the Digit 360 modality. In Fig. 2, the four tactile images are difficult to distinguish. Consider including the object’s CAD model and/or overlays highlighting contact regions to clarify how the signals correlate with surface geometry.

2. L139–140: ‘For instance, when manipulating an object in-hand, touch provides context about forces, slip, and subtle pose changes.’ It’s true that humans can interpret gel deformation or temporal changes to understand forces, slip, and subtle pose changes. Do the model or the current architecture actually capture this information? I would be interested to see further evidence that such physical information is leveraged by the model, instead of only reporting success rates over several simple tasks.

3. Would be interested to see the tactile signals for both t_k timestep and t_{k+1}. This could better help reader to understand how your model predicted based on the history.

4. Its hard to say reach button, push fruits, reach and push, stack cubes are tasks that require tactile information to complement the vision. More contact-rich tasks are appreciate.

5. Images in figure 5 are hard to recognize. It make reader hard to get the information the author want to carry out.

6. In figure 5, the author mentions in the vision only, the object less visible. Actually, its still less "visible" for both methods in the camera view. The author could show tactile signals for VT-WM, if it can be clearly reflected in tactile images, we can consider its more visible in tactile view.

7. Line 358-360. "This highlights the V-WM’s difficulty to distinguish between contact and non- contact  states  based  on  visual  input  alone." I agree the statement for vision-only. Would be appreciate what levels of contact information can digit 360 and the model could extract from the tactile signals, contact or not only, slip, shear force, normal force. A detailed description of tactile raw signals' performance and the model's capabilities to extract tactile information are appreciate.

Major weakness:

1. Would be better to specify the dataset used to train the world model in the main text. Eg. details in A.0.1 in Appendix. I still curious how large the dataset is and how many data pair(proprioception, vision, tactile) you used for the training. As the allegro hand generally is less stiff, a small control difference will largely change the tactile signals in fingertip( but the object may still be grasped in hand). I assume it may require large data to capture such physics between hand control and tactile.

2. It is hard to claim ‘data efficiency,’ since a large amount of data is required to train the world model and it includes the tasks used in the downstream evaluation.

3. Regarding the eight tasks in Appendix Fig. 9 used to train the world model, five of those tasks are used for final fine-tuning and for the results reported in the main text.  It would be better to exclude those tasks from world-model training and try more contact-rich tasks outside these eight to see whether it performs better or requires less data. Currently, I only see a new task. placing a plate in a dish rack, which is a simple pick-and-place and does not require tactile sensing.

4. When reporting the numbers in Figure 8, five of the eight training tasks are included. I assume the tasks ‘scribble with marker,’ ‘insert lampshade,’ and ‘insert table leg’ are more complex, which is understandable. Rather than avoiding report their results, I’m more curious about what limits these tasks from succeeding, or what should be added to make them work, for example, adding more data to train the world model or adding more data for fine-tuning.

5. It is actually unfair to compare a fine-tuned world model with a simple BC policy trained from scratch, since training the world model requires a lot of additional data, and the downstream tasks are also included in the world model. A more comparable baseline would be preferable. For example, using the same dataset to train a pretrained model [1], such as a simple cross-modal vision-and-tactile model, and then fine-tuning a state-of-the-art BC like diffusion policy or ACT. This would be fairer because it uses the same dataset for pretrain, rather than training BC from scratch.

[1] ViTaMIn: Learning Contact-Rich Tasks Through Robot-Free Visuo-Tactile Manipulation Interface.

6. Are there any baseline comparisons for the tactile encoder? For example, why use the Sparsh-X tokenizer? Baselines to justify the Cosmos tokenizer and the Sparsh-X tokenizer would be appreciated.

**Questions:**

See weakness above.

---

> ### Author Response · Authors · 2025-11-21
> **Response to Reviewer MMZP (part 1)**
>
> We sincerely thank you for your thorough review and for recognizing the strong presentation and clear reporting of our results. We address your major concerns in detail below, followed by responses to your other points. We added your visualization suggestions to the manuscript.
>
> ## Major Weakness
>
> 1. *Dataset size … it may require large data to capture such physics between hand control and tactile*
>
>     We agree that substantial data is needed to capture finger–object interactions in tactile representations. Because we use the pretrained Sparsh-X encoder, our model implicitly benefits from the large-scale tactile dataset used to train that [Sparsh-X](https://openreview.net/forum?id=sMs4pJYhWi#discussion).
>
>      Our own dataset for training V-WM and VT-WM contains 124 demonstrations (112k datapoints, ~40sec per demo). The validation set includes 26 demonstrations across all tasks, totaling 17k datapoints. We have added this information to Appendix A.
> &nbsp;
>
> 2. *It is hard to claim ‘data efficiency’*
>
>     The new task (placing the plate in the dish rack) is unseen during WM training, although its underlying skills (e.g., reaching, insertion) are within the training distribution. This is exactly what we aim to evaluate: if the WM already captures these primitives, then solving a novel task should require only a small number of demonstrations for fine-tuning. We agree that we operate in a low-data regime for pretraining the WM, the reason why some fine-tuning is still necessary.
> &nbsp;
>
> 3. *tasks are used for final fine-tuning and for the results reported in the main text. It would be better to exclude those tasks from world-model training and try more contact-rich tasks*
>
>     We clarify that for the planning evaluation on seen tasks, we do not fine-tune the WM. Although the task goal is in-distribution, for each trial, we randomize both the object’s initial position and the robot’s starting pose. This setup tests whether the WM maintains consistent action controllability and whether tactile input provides contact awareness that complements the kinematic information learned from vision.
> &nbsp;
>
> 4. *the tasks ‘scribble with marker,’ ‘insert lampshade,’ and ‘insert table leg’ … Rather than avoiding report their results, I’m more curious about what limits these tasks from succeeding, or what should be added to make them work,*
>
>     Tasks like “scribble with marker,” “insert lampshade,” and “insert table leg” are used solely for WM pre-training. We exclude them from planning evaluation because they require dexterous hand control, whereas our planning pipeline treats the hand as a gripper and is not designed to recover from complex errors. Handling such tasks would require a method for learning high-dimensional policies with the WM, which is an interesting direction but outside the scope of this work.
> &nbsp;
>
> 5. *unfair to compare a fine-tuned world model with a simple BC policy trained from scratch … a more comparable baseline would be using the same dataset to train a pretrained model [1], such as a simple cross-modal vision-and-tactile model*
>
>     This is an excellent point regarding the baseline comparison. Our goal with this comparison is to illustrate how a world model (WM) can support planning in tasks with few demonstrations, leveraging skills already embedded in the WM. Using single-task BC as a baseline helps highlight the advantages of pre-learning manipulation dynamics across many skills, compared to mimicking a specific behavior from scratch.
> &nbsp;
>
> 6. *Are there any baseline comparisons for the tactile encoder? … Baselines to justify the Cosmos tokenizer and the Sparsh-X tokenizer would be appreciated*
>
>     We acknowledge the interest in encoder ablations. In our work, we build on Cosmos and Sparsh-X, whose publications already provide extensive ablations and probing analyses demonstrating the richness of their learned representations. Following common practice in robot policy learning, we adopt these pre-trained encoders as reliable sources of visual and tactile latent features. This approach is consistent with prior influential works—for instance, [1] rely solely on CLIP encoders, [2] on PaliGemma VLM initialization, and [3] on CLIP embeddings for the vision modality.
>
>     [1] ViTaMIn: Learning Contact-Rich Tasks Through Robot-Free Visuo-Tactile Manipulation Interface
>     [2] A Careful Examination of Large Behavior Models for Multitask Dexterous Manipulation
>     [3] π0: A Vision-Language-Action Flow Model for General Robot Control

---

> > ### Author Response · Authors · 2025-11-21
> > **Response to Reviewer MMZP (part 2)**
> >
> > ## Minor Weakness
> >
> > 1. *Visualization of tactile signals:* We have added visualizations of the predicted tactile signals for multiple timesteps in Appendix B (Figs. 12–13). Specifically, Fig. 12 shows the predicted tactile signatures from VT-WM when rolling out the model with ground-truth actions, alongside the ground-truth tactile signals. We also highlight the difference from the no-contact state to make tactile imprints easier to interpret. Fig. 13 complements the results from Fig. 5, showing the tactile signatures imagined by VT-WM for the cube-stacking task, showing that the object is perceived in the tactile modality.
> > &nbsp;
> >
> > 2. *Relevance of contact-rich tasks:* While the tasks (e.g., reach button, push fruits, stack cubes) are simple, they already illustrate the benefits of a model that imagines tactile signatures. Vision alone can be ambiguous. For example, the exact moment of contact or object depth may be unclear. Tactile information disambiguates interactions, allowing VT-WM plans to be more accurate and successful in the real world.
> > &nbsp;
> >
> > 3. *Model capabilities in leveraging tactile signals:* As shown in the new visualizations (Figs. 12–13), VT-WM captures at least contact vs. no-contact states congruent with visual predictions. This confirms that the model leverages tactile information to complement vision, improving planning fidelity.

---

### Official Review · Reviewer_A4Jw · 2025-11-01

**Soundness:** 3
**Presentation:** 4
**Contribution:** 3
**Rating:** 6
**Confidence:** 4

**Summary:**

This paper introduces an action-conditioned world model that simultaneously takes visual and tactile inputs and predicts future visual and tactile observations. To demonstrate effectiveness, the authors conduct experiments to assess object permanence and causal compliance. Additionally, the authors apply the model to robotic manipulation and report data efficiency gains over behavior cloning.

**Strengths:**

* Novelty: To the best of my knowledge, this paper presents one of the first visuo-tactile world models.
* Thorough experiments: The paper includes experiments on video generation, robotic manipulation, and data efficiency. These comprehensive studies make the work quite solid.

**Weaknesses:**

* The explanation for causal compliance is not entirely satisfactory. As shown in Fig. 7, VT-WM appears to understand contact and predicts the cloth to be static. However, in theory, the model could also hallucinate incorrect tactile images and predict contact. Adding a tactile modality does not necessarily guarantee causal compliance. Based on Fig. 7, it is difficult to rule out cherry-picking, and the evidence does not fully establish causal compliance. For example, in Fig. 7, it seems plausible that VT-WM could imagine contact and predict object motion.
* The robot planning formulation is unclear. The authors claim the search space is R^7, covering the robot hand pose and gripper closure. First, this search space is still large; considering the planning horizon H, the problem remains challenging, and more details are needed on how the authors tackle it. Second, a dexterous hand has many more degrees of freedom than simple closing/opening. Reducing high-DoF dexterous control to a single DoF significantly simplifies the problem and may undermine the motivation for using a dexterous hand; a parallel-jaw gripper might suffice.
* The robotic tasks are relatively simple. This likely reflects a broader challenge for the community and may be outside this paper’s scope.
* Comparing VT-WM + planning to behavior cloning (BC) is questionable. The outcome depends heavily on task design. For challenging, long-horizon, contact-rich tasks, VT-WM + planning may not outperform BC. Therefore, the comparison offers limited insight because results are highly task-dependent and sensitive to design choices. Additionally, VT-WM + planning can leverage prior datasets, whereas BC cannot.

**Questions:**

* What is the inference time of the world model, and what is the speed of robot planning? Real-time performance is important for practical robotics.
* How consistent are the tactile and visual predictions? Is it possible for the imagined tactile signal to indicate contact while the visual prediction suggests otherwise?
* I may have missed this, but what is the control horizon for each robotic task?
* Could the authors provide more details on the CEM method, such as pseudocode?

---

> ### Author Response · Authors · 2025-11-21
> **Response to Reviewer A4Jw**
>
> Thank you for your detailed review and we appreciate that you recognize the novelty of introducing one of the first visuo-tactile world models (VT-WM), the thoroughness of our experiments, and the excellent presentation of the paper. We address your concerns and questions below. We have also updated the manuscript based on your suggestions.
>
> ### **Computational Performance**
>
> In our pipeline, robot plans are computed offline and applied in an open-loop manner. The key insight is that tactile is key for the prediction and planning of contact-rich tasks. Closed-loop execution is of great interest and for future work.
> Our timings on an RTX 3080 are:
>
> *World Model Inference:*
> - Autoregressive rollout for a $2$ second horizon takes $1.17$ seconds.
> - Including the encoding and decoding of observations and actions, the total time for a rollout is $1.97$ seconds.
> - Planning Time: each full CEM call ($36$ particles for $10$ iterations) takes approximately $66$ seconds.
>
> *Planning Time:* each full CEM call ($36$ particles for $10$ iterations) takes approximately $66$ seconds.
>
> ### **Visuo-Tactile Consistency**
>
> Yes, some mismatch can occur. However, we would like to highlight that our primary goal rather than generating perfect raw sensory prediction, is to build a world model with contact awareness. We are interested in studying  the advantages of providing physical grounding to world models, in this case via tactile sensing,  such that in general their utility for planning improves for contact-rich tasks.
>
> ### **Control Horizon and CEM Details**
>
> The core planning horizon for each CEM call is $2$ seconds with the WM generating at 6fps. We employ a re-planning strategy: we divide a task into subgoals, and for each subgoal, we call the CEM planner up to 3 times. This allows the system to reach the goal if the initial plan falls short or to actively recover from minor execution errors. The CEM parameters are 36 particles for 10 iterations.
>
> We provide here a compact pseudocode for CEM planning and added a detailed version in Appendix A.C.1.
>
> >  **CEM Planning with World Model**
> >
> >**Inputs:** WM (world model), goal image $X_{rgb}^{goal}$, context $X_{rgb}^0$, $X_{touch}^0$
> >
> >1.  **Encode goal and context:** [cite: 74]
> >    *  $Z^{goal} = \text{vision-encoder}(X_{rgb}^{goal})$
> >    *  $Z_{rgb}^0 = \text{vision-encoder}(X_{rgb}^0)$
> >    *  $Z_{touch}^0 = \text{touch-encoder}(X_{touch}^0)$
> >
> >2.  **Initialize CEM action distribution** ($\text{mean } \mu$, $\text{std } \sigma$)
> >
> >3.  **For each iteration** $(1..N)$:
> >    *  **a.** Sample action sequences from $N(\mu,\sigma)$
> >    *  **b.** Rollout actions in WM: $(Z_{rgb-pred}^{1:H}, Z_{touch-pred}^{1:H}) = \text{WM.rollout}(Z_{rgb}^0, Z_{touch}^0, \text{actions})$
> >    * **c.** Compute cost: $\text{costs} = L2(Z^{goal}, Z_{rgb-pred}^H)$
> >    * **d.** Select top-K elite actions and update $\mu$, $\sigma$
> >
> >4.  **Execute best action sequence on robot**
> >
>
> ### **Robot Planning Formulation, high-DoF simplification and tasks relatively simple**
>
> The 7 dimensions are indeed the 6-DoF end-effector pose and the 1-DoF gripper closure, making $\mathbb{R}^7$. We agree that this search space is still large. Nevertheless, the planning tasks do not fully depend on changing the rotation of the wrist, and the planner can exploit this. Works like [1], [2] have also used CEM for searching an action space with similar dimensionality
>
> It’s true that the reduction of the Allegro hand to a single DoF for planning is a simplification. However, this choice still allows us to validate that VT-WM's contact-grounded imagination can successfully drive an off-the-shelf planner (CEM) for contact-rich tasks.
>
> Although our tasks are simple tabletop manipulation, they are all contact rich (e.g., cube pushing, plate insertion, precise object alignment). Moreover, in comparison to prior work on WM for robot manipulation, our tasks go beyond moving in free space and reaching-like tasks.
>
> The main contribution is the WM's contact-grounded predictive power. We demonstrated that even with a simplified action space, the inclusion of touch enables better planning performance.
>
> [1] V-JEPA 2: Self-Supervised Video Models Enable Understanding, Prediction and Planning
> [2] DINO-WM: World Models on Pre-trained Visual Features enable Zero-shot Planning
>
> ### **BC comparison**
>
> We agree that the BC comparison is task dependent. For a new task, the popular approach in robotics would be to train a BC policy with the data for the new task. We wanted to get an intuition about how the VT-WM will do in terms of data efficiency, given that the new task contains skills that are already in-domain (e.g. reaching, pushing, inserting). That is why we restrict our comparison to a single task and we focus on studying performance given a limited budget of training data.

---

> > ### Comment · Reviewer_A4Jw · 2025-11-26
> >
> > Most of my questions are answered. I think this paper overall has a good quality, but slow CEM and control simplification make it less satisfactory. Therefore, I would like to keep my ratings.

---

### Author Response · Authors · 2025-11-21
**Summary of Revisions**

We sincerely thank all the reviewers (A4Jw, MMZP, KZLz, 8r8W) for their thoughtful and constructive feedback. The reviewers collectively acknowledged the core strengths of our submission:

- *Novelty and Importance (A4Jw, KZLz):* Recognizing the importance of the problem and our work as one of the first visuo-tactile world models (VT-WM).

- *Thoroughness (A4Jw, MMZP, KZLz, 8r8W):* Experiments are compelling and interesting (8r8W), results are clearly reposted (MMZP), highlighting the efforts on giving statistical significance (KZLz), which make the work "quite solid" (A4Jw).

- *Soundness & Presentation (MMZP, KZLz, 8r8W):* Acknowledging the paper's clear organization and excellent presentation.

In response to the reviewers' comments, we have updated the manuscript with all suggested clarifications, pseudocode, and visualizations. All revisions are highlighted in blue for easy identification.

The major concerns raised by the reviewers and our corresponding responses are summarized below.

1. **Data Efficiency Claim and Baseline Comparison (MMZP, KZLz).**  Reviewers questioned the fairness of the "data efficiency" comparison against a single-task BC policy for the new task (placing the plate in the dish rack). We chose to compare against a specialist policy (ACT) because, in practice, this represents the strongest possible baseline for that specific task.

    The new task is unseen during WM training, although its underlying skills (e.g., reaching, insertion) are within the training distribution. This is exactly what we aim to evaluate: if the WM already captures these primitives, then solving a novel task should require only a small number of demonstrations for fine-tuning.

    The $\mathbf{3.5x}$ improvement in success rate on the new task demonstrates that the generative, contact-grounded knowledge of the VT-WM is a significantly more powerful and data-efficient prior for planning and task transfer than the specialist policy alone.

2. **Tactile Modality Prediction and Causal Compliance (A4Jw, MMZP).** Reviewers questioned whether simple tasks justify the tactile modality and whether the model truly achieves "causal compliance".

    The value of touch is proven by statistical fidelity in imagination. Our quantitative metrics (Fig. 4, 6) provide the necessary evidence. VT-WM achieves 33% better object permanence and 29% better causal compliance in autoregressive rollouts than V-WM. This is statistical evidence that the tactile input successfully grounds the imagination in physical laws: object move when there is contact and a perturbation force.

    VT-WM is able to imagine which fingers are in contact during object interaction.This is crucial for avoiding failures like overshooting or hallucinating object movement without force. We addressed the request for more detailed visualization of tactile predictions by adding figures to Appendix B (Fig. 12 & 13), which clearly show the predicted tactile imprints during rollouts, visually confirming the model's utilization of contact information.

*Summary of Additional Manuscript Updates*

- CEM Pseudocode (A4Jw): Added detailed pseudocode for the Cross-Entropy Method (CEM) planning algorithm to Appendix C.

- Dataset size details (MMZP)

- Tactile predictions from VT-WM (MMZP): We added visualizations (Appendix B, Fig. 12 & 13)  illustrating the alignment between vision and tactile predictions in the VT-WM autoregressive rollout.

- Limitations Clarified (8r8W): We clarified that our touch modality is acquired from vision-based tactile sensors.

---

### Meta-Review · Area_Chair_5CXw · 2026-01-05

**Summary:**

The explanation for causal compliance is not entirely satisfactory.

The planning formulation is unclear.

The tasks are relatively simple.

Comparing VT-WM + planning to behavior cloning (BC) is questionable.

Slow CEM and control simplification.

Need additional visualization.

Clarity problems with verifications and methods.

Over-claim of data efficiency.

Figure 8 misses some tasks.

The tactile stream is limited to vision-based tactile images.

Most data comes from similar tabletop setups and Digit 360 configuration.

**Reviewer Concerns:**

The main clarity problems, the concerns about the planning, visualizations, and discussions, were addressed. However, the main concerns about the simple task design, insufficient result discussion, simple data recipe, and so on remained.

**Reviewer Scores:**

The positive reviewers hold their rating. I don't think the negative reviewers will increase the score, as the concerns about the task and experiments remain.

---

### Decision · Program_Chairs · 2026-01-26

Reject